# Crystal Plasticity Model Analysis of the Effect of Short-Range Order on Strength-Plasticity of Medium Entropy Alloys

Chen Li [1,2], Fuhua Cao [1,2], Yan Chen [1,2], Haiying Wang [1,2] and Lanhong Dai [1,2,*]

1   State Key Laboratory of Nonlinear Mechanics, Institute of Mechanics, Chinese Academy of Sciences, Beijing 100190, China
2   School of Engineering Science, University of Chinese Academy of Sciences, Beijing 101408, China
*   Correspondence: lhdai@lnm.imech.ac.cn

**Abstract:** Numerous studies have demonstrated the widespread presence of chemical short-range order (SRO) in medium and high entropy alloys (M/HEAs). However, the mechanism of their influence on macroscopic mechanical behavior remains to be understood. In this paper, we propose a novel dislocation-based model of crystal plasticity, by considering both the dislocation blocking and coplanar slip induced by SRO. The effect of SRO on the plastic deformation of CoCrNi MEAs was investigated. We found that the yield strength increases monotonically with increasing SRO-induced slip resistance, but the elongation first appeared to increase and then decreased. Further analysis suggested that the plastic elongation is a result of the competition between grain rotation-induced deformation coordination and stress concentration, which depends on the slip resistance of the SRO.

**Keywords:** short-range order; medium and high entropy alloys; crystal plasticity model

## 1. Introduction

Due to their remarkable mechanical properties, medium and high entropy alloys (M/HEAs), which include multiple primary elements in almost equal amounts, have recently received a lot of attention [1–13]. M/HEAs were initially conceived as random solid solutions, with a perfect chemical disorder organization. The mismatched atomic radii and intricate interactions between the constituent elements, however, provided evidence that the arrangement of atoms in M/HEAs is not entirely random [14–16]. Chemical elements can be rearranged to lower their free energy during solidification or/and heat treatment operations. This results in the formation of specific preferential inter-atomic neighborhoods and the production of chemical short-range orders (SRO) [17–20]. Accordingly, the presence of SROs may be a general, yet distinctive, structural characteristic of M/HEAs, and which can be used to adjust properties even more effectively.

From many atomic-level simulations and experimental studies, it is known that the SRO interacts with dislocations, the SRO impedes dislocation motion, and when sufficient dislocations pass through an SRO region (severe plastic deformation), the order degree of the SRO region decreases (towards a random solid solution) [18,21–23]. In addition, the increasing degree of SRO in an alloy material leads to a transformation of the dislocation slip, from a wavy to coplanar configuration [19,24–26]. Although in different ways, all these results highlight the strong influence of SRO on mechanical properties. However, due to the chemical and structural complexity of M/HEAs, understanding their full impact on mechanical behavior is challenging. Further efforts are needed, to characterize the extent and macroscopic processes that influence the mechanical behavior of M/HEAs through the SRO mechanism.

In this paper, a modified crystal plasticity finite element model is developed to analyze the influence of SRO on deformation behaviors, in which the effects of both dislocation impedance and coplanar slip by SRO are considered. The typical CoCrNi MEAs, in which SRO is well observed in both experiments and simulations, was taken as the model

alloy [4,19,22]. The modified model was used to simulate a tensile experiment of the polycrystalline CoCrNi MEAs and confirm its corresponding parameters. On this basis, the effect of deformation is discussed, by adjusting the degree of SRO.

## 2. Crystal Plasticity Framework

### 2.1. Kinematics of Crystal Plasticity

The total deformation gradient tensor **F** (bold font indicates second-order tensor) can be decomposed as:

$$\mathbf{F} = \mathbf{F}^* \cdot \mathbf{F}^P \tag{1}$$

where the superscript $*$ denotes lattice deformation and superscript $P$ denotes plasticity. The total velocity gradient is defined as [27]:

$$\mathbf{L} = \dot{\mathbf{F}} \cdot \mathbf{F}^{-1} = \mathbf{L}^* + \mathbf{L}^P \tag{2}$$

The total velocity gradient can be decomposed into the plasticity part $\mathbf{L}^P$ and the lattice deformation part $\mathbf{L}^*$. The strain rate $D^*$ is the symmetric part of $\mathbf{L}^*$. Using the Jaumann objective stress rate for Kirchhoff stress, the instanton equation is expressed as:

$$\overset{\nabla}{\boldsymbol{\tau}} = \left[ (1 - f_{tw})\mathbb{C}_m + \sum_{\beta=1}^{N_{tw}} f^\beta \mathbb{C}_{tw}^\beta \right] : \mathbf{D}^* \tag{3}$$

where the stiffness matrix is considered to be a linear combination of the stiffness matrix of the matrix and the twin, Kirchhoff stress $\boldsymbol{\tau} = J\boldsymbol{\sigma}$, $J = \det(\mathbf{F})$, $\boldsymbol{\sigma}$ is the Cauchy stress, $f_{tw} = \sum_{\beta=1}^{N_{tw}} f^\beta$ is the total volume of the twin, $f^\beta$ (regular italic font indicates scalar) and $\mathbb{C}_{tw}^\beta$ (Euclid math font indicates fourth-order tensor) are the volume fraction and the stiffness of twins for the twin system $\beta$, respectively. $\mathbb{C}_m$ is the stiffness of the matrix material. $N_{tw}$ is the total number of twin systems.

The velocity gradient of plasticity consists of dislocation slip and twinning:

$$\mathbf{L}^P = \left( 1 - \sum_{\beta=1}^{N_{tw}} f^\beta \right) \sum_{\alpha=1}^{N_s} \dot{\gamma}^\alpha \boldsymbol{m}^\alpha \otimes \boldsymbol{n}^\alpha + \sum_{\beta=1}^{N_{tw}} \dot{\gamma}^\beta \boldsymbol{m}_{tw}^\beta \otimes \boldsymbol{n}_{tw}^\beta \tag{4}$$

where $N_s$ is the total number of slip systems. $\dot{\gamma}^\alpha$ and $\dot{\gamma}^\beta$ are the individual shear strain rates of the slip system $\alpha$ and twin system $\beta$. $\boldsymbol{m}^\alpha$ and $\boldsymbol{n}^\alpha$ (bold italic font indicates vector (first-order tensor)) are the unit vectors along the shear and normal directions of the slip plane, while $\boldsymbol{m}_{tw}^\beta$ and $\boldsymbol{n}_{tw}^\beta$ are those of the twin plane.

### 2.2. Dislocation Slipping

In a dislocation-based model, the Orowan equation [28] gives the shear rate on the slip system $\alpha$ as:

$$\dot{\gamma}^\alpha = \rho^\alpha b v_0 \exp \left\{ -\frac{Q_s}{k_B T} \left[ 1 - \left( \frac{\tau_{eff}^\alpha}{\tau_{sol}} \right)^p \right]^q \right\} \mathrm{sign}(\tau^\alpha) \tag{5}$$

where $\rho^\alpha$ and $\tau_{eff}^\alpha$ are the dislocation density and the effective resolved shear stress on the slip system $\alpha$, $\tau_{sol}$ is the solid solution strength, $b$ is the length of the Burgers vector for slip, $v_0$ is the dislocation glide velocity, $Q_s$ is the activation energy for dislocation slip, $k_B$ is the Boltzmann constant, $T$ is the temperature, and $p$ and $q$ are fitting parameters.

The effective shear stress $\tau_{eff}^\alpha$ is modified on the basis of Xiaochong Lu [29]:

$$\tau_{eff}^\alpha = |\tau^\alpha - \tau_b^\alpha| - \tau_p^\alpha - \tau_{SRO}^\alpha \tag{6}$$

where $\tau^\alpha$ is the corresponding resolved shear stress of the slip system $\alpha$, $\tau_b^\alpha$ is the back stress, $\tau_p^\alpha$ is caused by the obstacles of forest dislocation and Hall-Petch stress, and $\tau_{SRO}^\alpha$ is dislocation resistance of SRO. The regions of SRO can be thought of as a smaller-sized and more densely distributed eutectic lattice precipitation [16,19], which has a certain pegging effect on the dislocation motion and can improve the yield strength at the macroscopic level [22]. Therefore, adding a negative term to the effective stress for the resistance of SRO increases the yield strength and decreases the slip velocity in the slip system.

The first term $\tau^\alpha$ in the right of Equation (6) is the resolved shear stress is [27]:

$$\tau^\alpha = \boldsymbol{m}^\alpha \cdot J\boldsymbol{\sigma} \cdot \boldsymbol{n}^\alpha \tag{7}$$

The second term $\tau_b^\alpha$ in the right of Equation (6) is the back stress, which is caused by the dislocation pile-ups at the grain and twin boundaries [28]:

$$\tau_b^\alpha = \frac{Gb}{\Lambda_s^\alpha} N^\alpha \left( 1 - \frac{N^\alpha}{N^*} \right) \tag{8}$$

$$\dot{N}^\alpha = \frac{l}{b} \left( 1 - \left| \frac{N^\alpha}{N^*} \right| \right) \dot{\gamma}^\alpha \tag{9}$$

where $G$ is the shear modulus, $\Lambda_s^\alpha$ is the geometrical length scale of the microstructure, $l$ is the average distance between slip bands, $N^\alpha$ is the number of piled-up dislocations at boundaries, and $N^*$ is the saturated number of $N^\alpha$.

The microstructural geometrical length $\Lambda_s^\alpha$ in Equation (8) is defined by the combined effect of the grain and twin boundaries [29]:

$$\frac{1}{\Lambda_s^\alpha} = \frac{1}{d_{grain}} + \frac{1}{\lambda_{s-t}^\alpha} \tag{10}$$

$$\frac{1}{\lambda_{s-t}^\alpha} = \sum_{\beta=1}^{N_s} \xi_{\alpha\beta} f^\beta \frac{1}{t(1 - f_{tw})} \tag{11}$$

where $d_{grain}$ is the grain size, $\xi_{\alpha\beta}$ characterizes the interaction between the slip system $\alpha$ and twin system $\beta$, and $t$ is the average thickness of twin lamellas.

The passing stress $\tau_p^\alpha$ in Equation (6) is caused by the obstacles of forest dislocation and Hall–Petch stress [29]:

$$\tau_p^\alpha = \sigma_0 + \frac{k_{HP}}{\sqrt{d_{grain}}} + Gb \sqrt{\sum_{\alpha'=1}^{N_s} \xi_{\alpha\alpha'} \rho^{\alpha'}} \tag{12}$$

where $\sigma_0$, $k_{HP}$ are coefficients of the Hall–Petch law, and $\xi_{\alpha\alpha'}$ is the interaction coefficient between the slip system $\alpha$ and $\alpha'$, including the self-hardening, coplanar, collinear, orthogonal, glissile, and sessile interactions [30]. According to the modified Kocks–Mecking rule, the evolution rate of dislocation density can be expressed by [31]:

$$\dot{\rho}^\alpha = \left( \frac{1 - N^\alpha/N^*}{b \cdot \Lambda_s^\alpha} + \frac{k}{b} \sqrt{\rho^\alpha} - \frac{2d_{anni}}{b} \rho^\alpha \right) \left| \dot{\gamma}^\alpha \right| \tag{13}$$

where $k$ is the forest dislocation hardening constant, and $d_{anni}$ is the annihilation distance for dislocations.

In order to describe the dislocation blocking and coplanar slip induced by SRO, $\tau_{SRO}^\alpha$ can be decomposed into two terms: one for the initial resistance $\tau_{SRO}^0$, and one for the evolution coefficient $D_{SRO}^\alpha$. The evolution coefficient $D_{SRO}^\alpha$ decreases rapidly with

cumulative slip, and since the coefficient $D_{SRO}^{\alpha}$ should tend to zero, as the cumulative slip tends to infinity, an exponential form is used. The specific form is as follows:

$$\tau_{SRO}^{\alpha} = \tau_{SRO}^{0} D_{SRO}^{\alpha} = \tau_{SRO}^{0} \exp\left(-\frac{\sum\limits_{\eta=1}^{N_s} \xi_{\alpha\eta} \gamma_{cum}^{\eta}}{\gamma_0}\right) \tag{14}$$

where $\tau_{SRO}^{0}$ is the initial value of $\tau_{SRO}^{\alpha}$. $\gamma_0$ is the reference strain. $\xi_{\alpha\eta}$ is the dislocation resistance coefficient of SRO between the slip system $\alpha$ and $\eta$, including coplanar and noncoplanar interactions.

### 2.3. Deformation Twinning

Twining causes plastic deformation and serves as a grain-refining process. The twinning evolution adopts a similar form to that of Xiaochong Lu [29]. The equation of twinning evolution is briefly explained in this section. The twin system's contribution to the shear rate can be represented as follows:

$$\dot{\gamma}^{\beta} = \frac{\pi}{4} \gamma_{tw} (1 - f_t) \Lambda_t^{\beta} wt \dot{N}_t^{\beta} \tag{15}$$

where $\gamma_{tw}$ is the characteristic twinning shear strain as $\sqrt{2}/2$ for FCC (Face Center Cubic) lattice. $\Lambda_t^{\beta}$ is the mean spacing between two obstacles caused by deformation twins. $w$ is the available width of twin lamellas, when $w > \Lambda_t^{\beta}$, $w = \Lambda_t^{\beta}$. The addition of $w$ results from the fact that twins usually do not penetrate the entire grain when the grain size is large.

The equation of $\Lambda_t^{\beta}$ is similar to Equations (10) and (11):

$$\frac{1}{\Lambda_s^{\beta}} = \frac{1}{d_{grain}} + \frac{1}{\lambda_{s-t}^{\beta}} \tag{16}$$

$$\frac{1}{\lambda_{s-t}^{\beta'}} = \sum_{\beta=1}^{N_s} \xi_{\beta\beta'} f^{\beta} \frac{1}{t(1 - f_{tw})} \tag{17}$$

where $\xi_{\beta\beta'}$ is the interaction coefficient between the twin systems $\beta$ and $\beta'$. The twin nucleation rate $\dot{N}_t^{\beta}$ in Equation (15) is influenced by the dislocation cross-slip and the resolved shear stresses on twin systems:

$$\dot{N}_t^{\beta} = \dot{N}_0 p_{ncs} p_{tn} \tag{18}$$

where $\dot{N}_0$ is the reference twin nucleation rate. $p_{tn}$ is the probability of forming twins. The probability that no cross-slip occurs $p_{ncs}$ is calculated by:

$$p_{ncs} = 1 - \exp\left[-\frac{V_{cs}}{k_B T}\left(\tau_r - \tau^{\beta}\right)\right] \tag{19}$$

where $V_{cs}$ is the activation volume for dislocation cross-slip, $\tau_r$ is the stress needed to bring two partial dislocations from the equilibrium distance $x_0$ to the critical distance $x_c$ in which the twin nucleation can be facilitated. $\tau_r$ is expressed as:

$$\tau_r = \frac{Gb}{2\pi}\left(\frac{1}{x_0 + x_c} + \frac{1}{2x_0}\right) \tag{20}$$

The equilibrium distance $x_0$ between two partial dislocations is determined by:

$$x_0 = \frac{G}{\Gamma_{sf}} \frac{b_p^2}{8\pi} \frac{2+\nu}{1-\nu} \tag{21}$$

where $\Gamma_{sf}$ is the value of stacking fault energy and $\nu$ is the Poisson ratio. $b_p$ is the length of the Burgers vector of Shockley partial dislocation, and $\tau^\beta$ is the resolved shear stress on the twin system $\beta$:

$$\tau^\beta = \boldsymbol{m}_{tw}^\beta \cdot J\boldsymbol{\sigma} \cdot \boldsymbol{n}_{tw}^\beta \tag{22}$$

The resolved shear stress on a twin plane affects the twin nucleation rate. The probability of forming a twin is calculated by:

$$p_{tn} = \exp\left[-\left(\frac{\widehat{\tau}_t}{\tau^\beta}\right)^A\right] \tag{23}$$

where $A$ is a fitting parameter. $\widehat{\tau}_t$ is the critical stress for twinning, equal to the activation stress for a Frank–Read dislocation source plus the critical stress for generating a wide stacking fault:

$$\widehat{\tau}_t = \widehat{\tau}_{ds} + \widehat{\tau}_{sf} = \frac{Gb_p}{d_{grain}} + \frac{\Gamma_{sf}}{b_p} \tag{24}$$

In summary, using a dislocation-based crystal plasticity model that considers back stress, forest dislocation hardening, and twinning, the model introduces the impacts of SRO. The effect of the SRO impeding dislocation motion is reflected by adding $\tau_{SRO}^\alpha$ to the Orowan equation (Equations (5) and (6)), $\tau_{SRO}^\alpha$ decreases $\tau_{eff}^\alpha$, causing the dislocation slip velocity to decrease. Since the impeding effect of SRO varies with the slip of the material [22], the evolutionary coefficient $D_{SRO}^\alpha$ of $\tau_{SRO}^\alpha$ is given. Meanwhile, the interaction coefficient of the coplanar slip system is positive and the interaction coefficient of the non-coplanar slip system is negative, making the coplanar slip occurs easily, while the non-coplanar slip is suppressed.

## 3. Simulations and Validation of the Constitutive Model

### 3.1. Polycrystalline Finite Element Model

Simulation of polycrystalline CoCrNi MEAs used ABAQUS software (2020, Dassault Systemes Simulia Crop., Johnston, RI, USA). The material model of Section 2 was implemented using the Umat subroutine. A bone-shaped flat tensile sample is shown in Figure 1a. To reduce the calculation, an equivalent polycrystalline model was used for the clamping end (white part), and a crystal plasticity model was used for the samples in the parallel part, with different colors representing different grains (Figure 1a). The C3D8 (8-node linear brick) element was used in the simulation, with an element size of ~1 μm. The parallel segment size was $16 \times 40 \times 100$ μm$^3$ and contained a total of 55 grains (55 random points in parallel segment space generate 55 Voronoi polyhedra), so the grain size was about 13 μm.

Three groups of terms used in the paper are clarified prior to the analysis. The first group consists of macroscopic strain and local strain; the macro-strain is defined as the difference in displacement below and above the parallel segment divided by the length of the parallel segment, denoted by $E$; and the local strain is defined as integral point strain, denoted by $\varepsilon$. The second group consists of macroscopic stress and local stress; the macroscopic stress is equal to the sum of the restraint reaction forces divided by the cross-sectional area of the parallel section; and local stress is the stress of the integral point. The third group is the elongation, which is the macro-strain that corresponds to the highest point on the macro-stress. vs. macro-strain curve.

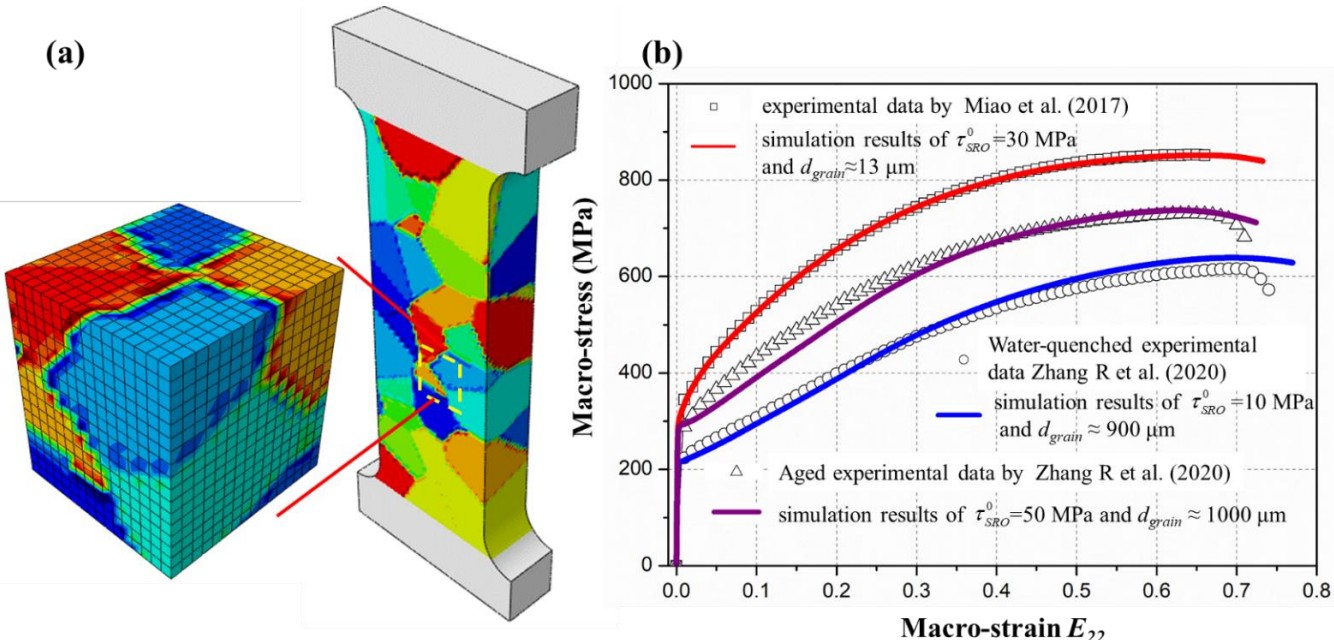

**Figure 1.** (**a**) Finite element model, the parallel part is colored according to the grain number. (**b**) Comparing simulation results of the developed model with experiments by Miao et al. [32] and Zhang R et al. [19].

### 3.2. Parameter Validation

To verify the validation of the developed constitutive model, the tensile responses of CoCrNi MEAs [32] were simulated in this section. The material parameters can be divided into three categories: the first category is the parameters that can be clarified, such as the lattice parameter, stacking fault energy [33], elastic properties [34], and dislocation activation energy [35]. The second category is parameters with ambiguous reference values, such as the average distance between slip bands, interaction coefficient between the twin systems, solid solution stress, the thickness of twin lamellas, etc. The values of these parameters were adjusted based on the parameters of TWIP (twinning induced plasticity) steel [29], which is similar to CoCrNi MEAs. Hall–Petch coefficients obtained in tensile experiments needed to be divided by the Taylor factor and adjusted appropriately according to the simulation results. The third category is the fitting parameters, such as, *p* and *q* Equation (5), *A* in Equation (23). In addition, the parameters related to SRO were also found by fitting with the experiment data. The specific parameters are shown in Table 1, and the fitting results is the red curve in Figure 1b, which fits well with the experimental results of Miao et al. [32].

To further confirm the feasibility of these parameters, the feature length of the model was adjusted from 1 μm to 69 μm and 77 μm (the average grain size from 13 μm to 900 μm and 1000 μm) to compare with the results in Zhang R and Zhao S [19] (the true stress–strain curve was transformed into engineering stress–strain curves, while taking the strain rate as $4 \times 10^{-3}$ s$^{-1}$). The blue curve corresponds to the curve of low SRO ($\tau_{SRO}^0 = 10$ MPa), and the purple curve is the curve of high SRO ($\tau_{SRO}^0 = 50$ MPa) (Figure 1b). These calculated results and the experiments match well, which shows the applicability of the parameters to CoCrNi MEAs.

**Table 1.** Material parameters used in the simulations.

| Symbol | Physical Mean | Value |
|---|---|---|
| $C_{11}$, $C_{12}$, $C_{44}$ | Elastic constants | 249, 156, 142 GPa |
| $\tau_{sol}$ | Solid solution strength | 200 MPa |
| $N_s$ | Total number of slip systems | 12 |
| $b$ | Burgers vector | 0.2522 nm |
| $N^*$ | Saturated number of piled-up dislocation | 39 |
| $l$ | Mean spacing between slip bands | 223 nm |
| $\sigma_0$, $k_{HP}$ | Hall–Petch coefficient (Converted to resolved shear stress) | 20 MPa, 88 MPa · $\mu$m$^{1/2}$ |
| $k$ | Forest dislocation hardening constant | 0.0488 |
| $\rho_0^\alpha$ | The initial dislocation density of the slip system | 4 $\mu$m$^{-2}$ |
| $v_0$ | Reference velocity for dislocation slip | $2 \times 10^{-4}$ m/s |
| $Q_s$ | The activation energy for dislocation slip | 0.27 eV |
| $p, q$ | The exponent in slip velocity | 0.75, 2.5 |
| $d_{anni}$ | Annihilation distance for dislocations | 1.1 $b$ |
| $\zeta_{\alpha\alpha'}$ | Interaction coefficient between slip systems | 0.122, 0.122, 0.625, 0.07, 0.137, 0.122 |
| $N_{tw}$ | Total number of twin systems | 12 |
| $h, t$ | The width and thickness of twin lamellas | 10 $\mu$m, 0.01 $\mu$m |
| $f_{max}^\beta$ | Maximum twin fraction of twin system | 0.01 |
| $V_{cs}$ | Cross-slip volume | 0.0469 nm$^3$ |
| $A$ | Twinning transition profile width exponent | 5 |
| $\dot{N}_0$ | reference twin nucleation rate | 2 s$^{-1}$ |
| $\zeta_{\alpha\beta}$ | Interaction coefficient between slip and twin systems | 0.0 (coplanar) 0.042 (cross-slip) |
| $\zeta_{\beta\beta'}$ | Interaction coefficient between twin systems | 0.0 (coplanar) 0.468 (non-coplanar) |
| $\Gamma_{sf}$ | Stacking fault energy | 22 mJ/m$^2$ |
| $\tau_{SRO}^0$ | Dislocation resistance of SRO | 10, 30, 50 MPa |
| $\gamma_0$ | reference strain | 0.25 |
| $\zeta_{\alpha\eta}$ | Interaction coefficient between slip and degree of SRO | 3 (coplanar) −1 (non-coplanar) |

## 4. Influence of SRO on Deformation Behavior

Based on the model developed in Section 2 and the parameters in Section 3, the effect of the resistance of SRO on the deformation behavior was evaluated by adjusting the value $\tau_{SRO}^0$. Figure 2a,b show the macro-stress vs. macro-strain curves at different $\tau_{SRO}^0$. By observing the curves, it can be seen that the yield strength increases monotonically when $\tau_{SRO}^0$ increases (Figure 2(a$_1$,b$_1$)). Furthermore, when observing the end of the curves, the elongation of the material rises and then falls as $\tau_{SRO}^0$ increases (Figure 2(a$_2$,b$_2$)). The observed phenomenon is plotted in Figure 2c. From Figure 2c, when $\tau_{SRO}^0 \approx 29$ MPa, the elongation takes the maximum value.

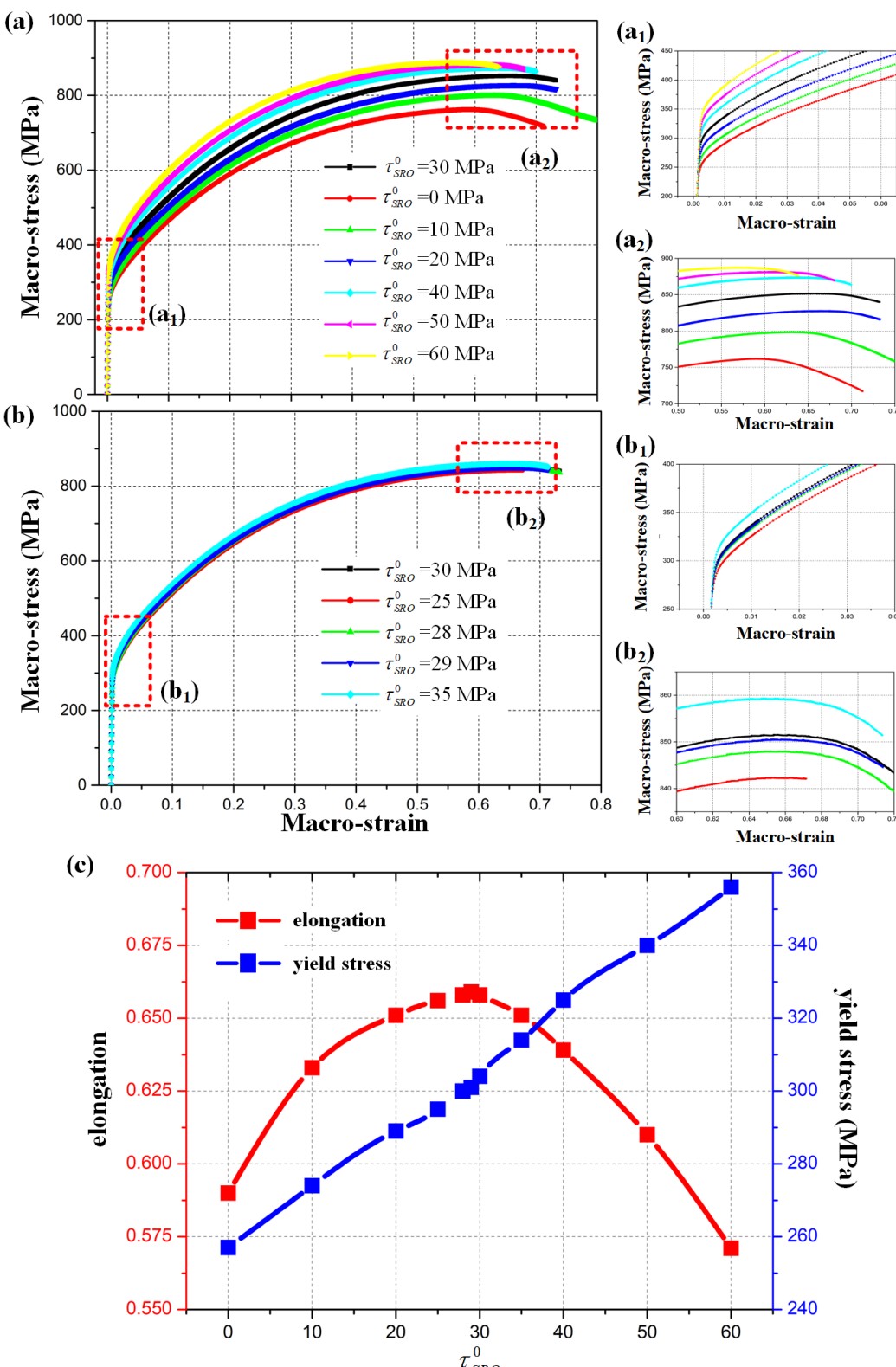

**Figure 2.** Macro stress–strain curves at different $\tau_{SRO}^0$. (**a**) $\tau_{SRO}^0$ are equal to 0, 10~60 MPa, respectively, the local enlargements for the head (**a$_1$**) and the end (**a$_2$**); (**b**) $\tau_{SRO}^0$ is equal to 25, 28, 29, 35 MPa, the values are near 30 MPa, the local enlargements for the head (**b$_1$**) and the end (**b$_2$**); (**c**) the curve of yield strength (Corresponding to the blue y-axis on the right) and elongation (Corresponding to the red y-axis on the left) with $\tau_{SRO}^0$.

To analyze why the elongation increases and then decreases with increasing $\tau^0_{SRO}$, we plotted the pole figures at $\tau^0_{SRO}$ = 0, 30, 60 MPa and macro-strains $E_{22}$ = 0.1, 0.3 (Figure 3). Figure 3a shows the distribution of 55 grains before deformation. As the micro-strain increases, the crystallographic orientation on the integration points unfold on the pole figures (Figure 3b–d). When $E_{22}$ = 0.1, the orientations of the integration points from grain #2 are dispersed on the pole figures (the red points in (Figure 3($b_1$,$c_1$,$d_1$)). When $E_{22}$ = 0.1 and $\tau^0_{SRO}$ = 0 MPa, the integration points from grain #2 are scattered in the region $\Delta l_0$ = 0.077 and $\Delta \theta_0$ = 11.17° (Figure 3($b_1$)). Similarly, when $E_{22}$ = 0.1 and $\tau^0_{SRO}$ = 30 MPa, the integration points are scattered in $\Delta l_{30}$ = 0.083 and $\Delta \theta_{30}$ = 12.78° (Figure 3($c_1$)). Moreover, when $E_{22}$ = 0.1 and $\tau^0_{SRO}$ = 60 MPa, $\Delta l_{60}$ = 0.099, $\Delta \theta_{60}$ = 13.12° (Figure 3($d_1$)). The crystallographic orientation of the material changes drastically as $\tau^0_{SRO}$ increases. It can be further inferred that the increase in elongation with increasing $\tau^0_{SRO}$, when $\tau^0_{SRO}$ < 29 MPa, is due to the intense local rotation, which can provide additional macro-strain.

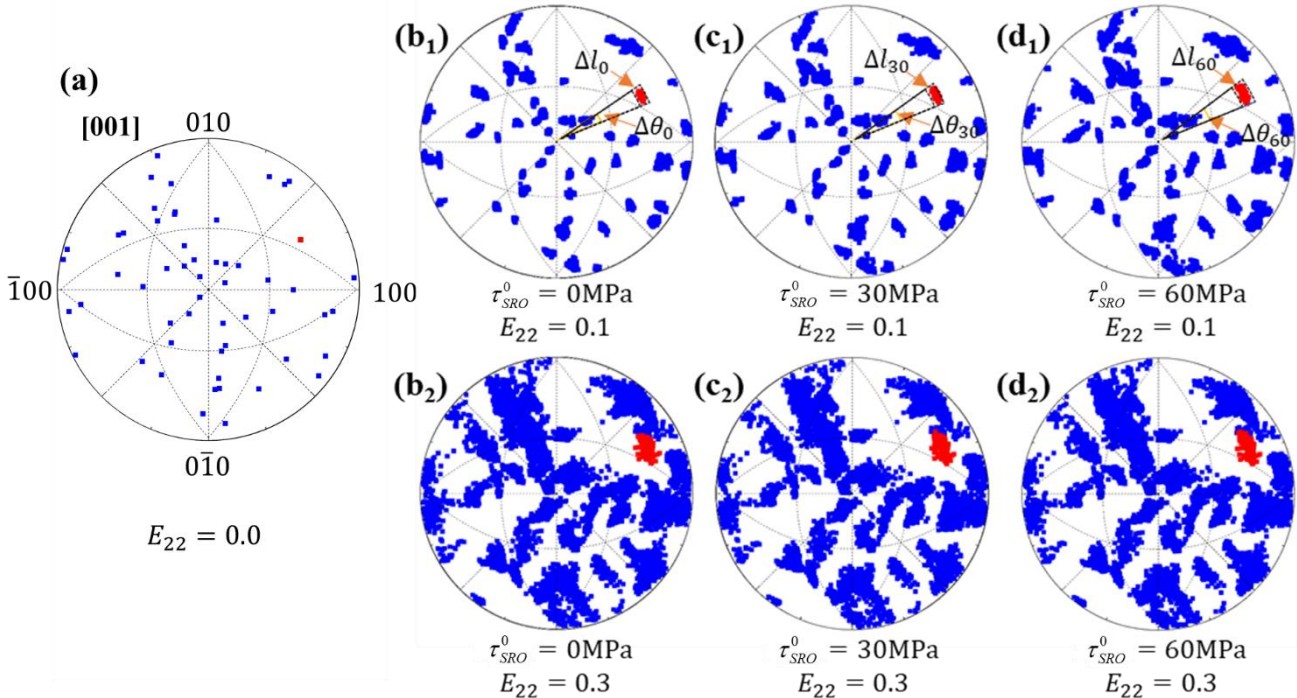

**Figure 3.** Pole figures at different $\tau^0_{SRO}$ and $E_{22}$. (**a**) Initial grain orientation distribution. (**$b_1$,$b_2$**) when $\tau^0_{SRO}$ = 0 MPa, the orientation distribution at $E_{22}$ = 0.1, 0.3. (**$c_1$,$c_2$**) when $\tau^0_{SRO}$ = 30 MPa, the orientation distribution at $E_{22}$ = 0.1, 0.3. (**$d_1$,$d_2$**) when $\tau^0_{SRO}$ = 60 MPa, the orientation distribution at $E_{22}$ = 0.1, 0.3. The red part is the orientation distribution of grain #2.

To illustrate the effect of local crystal rotation on macroscopic deformation, we chose a single crystal with $\varphi_1 \approx 9.74°$, $\Phi = 45°$, and $\varphi_2 = 0°$, for the simulation (Figure 4a). The grain was in the following orientation: the slip plane {111} was 45° to the tensile direction, and the slip plane {111} was easily activated, with a clear coplanar slip effect under the effect of SRO. As shown in (Figure 4b), the blue curve ($\tau^0_{SRO}$ = 0 MPa) and the red curve ($\tau^0_{SRO}$ = 30 MPa), both below the diagonal line (the black), indicate that the macro-strain was not only contributed to by the local (element) deformation, the rotation of the elements also provided a certain amount of macro-strain. When the macro-strains $E_{22}$ = 0.25, the mesh rotations in the central region of the specimen were compared, when $\tau^0_{SRO}$ = 0 MPa, the rotation angle $\theta^{single}_0 \approx 4.5°$ and when $\tau^0_{SRO}$ = 30 MPa, $\theta^{single}_{30} \approx 6.2°$. It follows that, when $\tau^0_{SRO}$ is large, the local rotation is also large, which increases the local rotation's contribution to the macro strain.

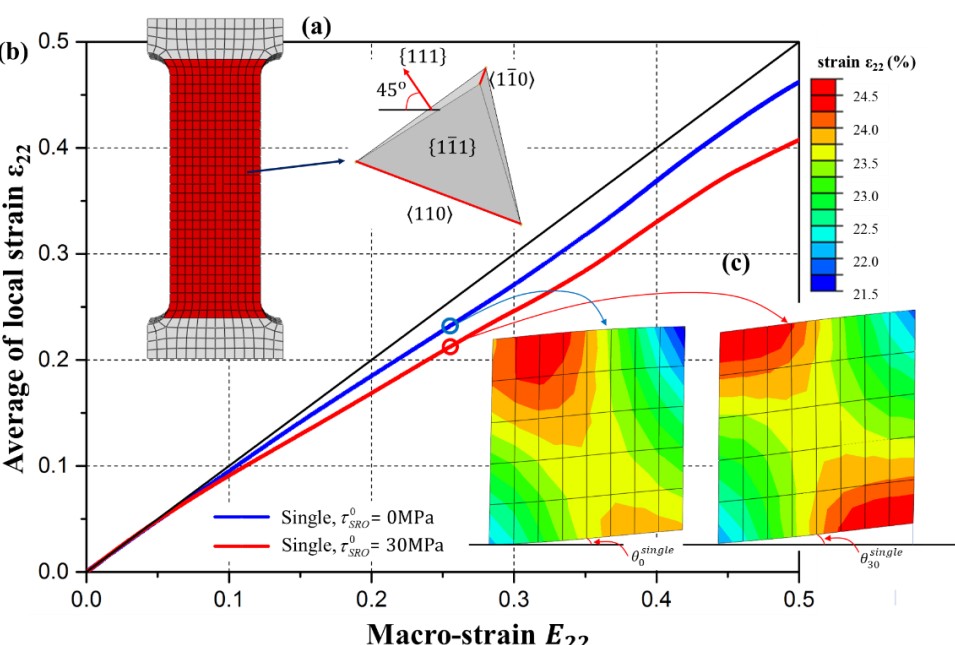

**Figure 4.** (**a**) Model of single crystal tensile simulation; the parallel section for the developed model, grain orientation as shown by the tetrahedron enclosed by the slip plane, {111} plane is 45° to the tensile direction, $\langle 1\bar{1}0 \rangle$ direction is perpendicular to the paper surface. (**b**) Comparison curves of macro-strain and average of local strain using single crystal tensile simulation at different $\tau_{SRO}^0 = 30$ MPa. (mesh size ~ 4 μm). (**c**) Comparison of mesh deformation under different $\tau_{SRO}^0$ for $E_{22} = 0.1$.

As seen from the results in Figure 4, the local rotation of the material becomes more intense when $\tau_{SRO}^0$ is larger. In polycrystalline materials, due to the different rotational directions generated by different grains, a more intense rotation causes a stronger stress concentration (Figure 5($a_1$,$b_1$,$c_1$)). From Equation (12), it is known that high local stress implies a high dislocation density; thus, quickly attaining a saturated dislocation density locally and causing the material to lose its capacity to harden (Figure 5($a_2$,$b_2$,$c_2$)).

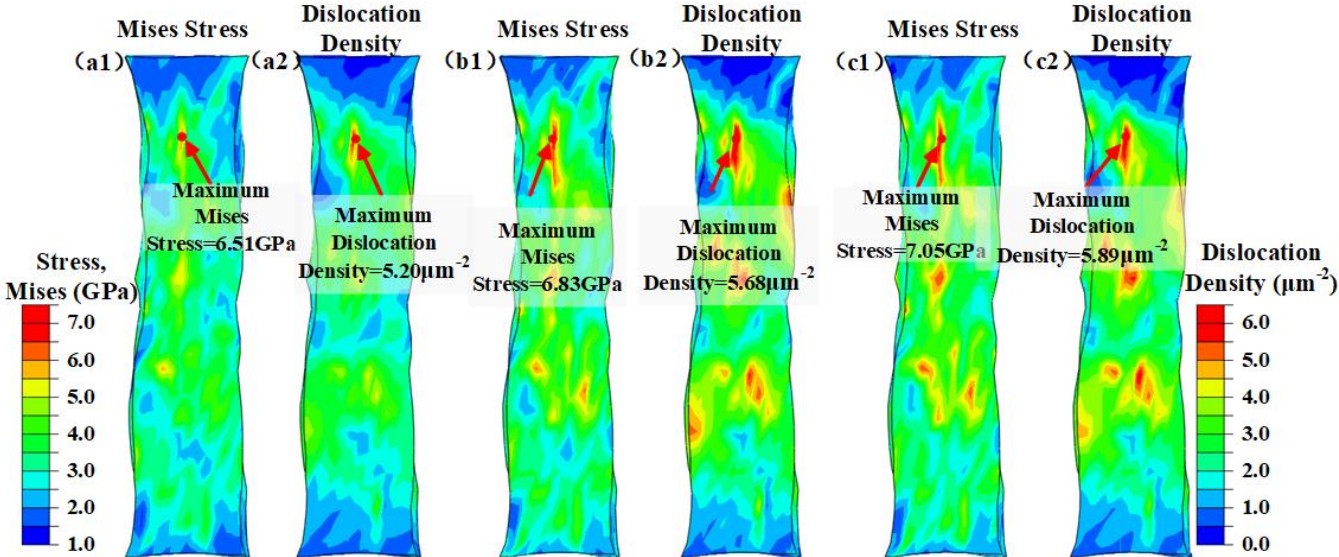

**Figure 5.** Mises stress and dislocation density distribution next to (**$a_1$,$a_2$**) $\tau_{SRO}^0 = 0$ MPa, (**$b_1$,$b_2$**) $\tau_{SRO}^0 = 30$ MPa, (**$c_1$,$c_2$**) $\tau_{SRO}^0 = 60$ MPa, with the $E_{22} = 0.5$.

The impact of SRO on the material's macroscopic deformation was explored using the developed model. The yield strength of the material rises monotonically as the amount increases, yet the elongation exhibits a trend of rising and then falling. The rise in dislocation resistance is the cause of the rise in yield strength. The intense grain rotation brought on by coplanar slip is the source of the initial rise and subsequent fall in elongation. When the $\tau_{SRO}^0 < 29$ MPa, the local grain rotation causes additional macro strain; while when the $\tau_{SRO}^0 > 29$ MPa, the intense rotation causes severe stress concentration, resulting in a decrease in elongation. There is a competitive relationship between the contribution of grain rotation to elongation and the decrease in elongation due to the stress concentration. When the resistance of SRO is small, this contribution dominates, and when the resistance is large, the degradation of stress concentration dominates. This model is a good explanation for the fact that the SRO causes non-uniform dislocation motion, which leads to plasticity improvement. On the other hand, with the increase of the order degree or density of SRO, the dislocation plug is aggravated during the deformation, which reduces the deformation ability of M/HEAs [16,19]. The monotonic increase in strength predicted by the model is consistent with the experimental results of Zhang R et al. [19], Ding et al. [16], and D. Han [24]. However, the results of elongation require more precise control of the SRO in experiments. The results of the simulation of Zhang R et al. [19], to a certain extent, show that the elongation decreases when the SRO is outside of the appropriate range.

## 5. Conclusions

Through the incorporation of the resistance of SRO into the dislocations and the coplanar slip effect into the dislocation-based model, this study offers a crystal plasticity model that is better suited to the M/HEAs. The main conclusions are as follows:

(1)　A set of parameters consistent with CoCrNi MEAs was determined and can be used to discuss the influence of various factors on a material's deformation behavior.

(2)　Adjusting the resistance of SRO at a certain range increases both the yield strength and elongation simultaneously, but beyond this range, the yield strength increases but the elongation decreases.

(3)　As the resistance of SRO increases, the elongation increases and then decreases, which is attributed to the more intense local rotation with coplanar slip. Local rotation can increase the additional macro strain, while also causing a more intense stress concentration; when the resistance of SRO is low, the additional macro strain dominates the elongation increase; when the resistance is high, the stress concentration dominates the elongation decrease.

The results predicted by the model indicate that the effect of the SRO on the uniform deformation of the M/HEAs is not monotonic; therefore, in the subsequent material design and treatment, whether the SRO is regulated by annealing treatment, or by changing or adding elements, the SRO should be controlled within a suitable range, to achieve a simultaneous improvement of strength and plastic deformation.

**Author Contributions:** Conceptualization, C.L., F.C. and L.D.; methodology, C.L., F.C. and Y.C.; software, C.L. and Y.C.; validation, C.L.; formal analysis, C.L.; investigation, C.L. and F.C.; resources, C.L.; writing—original draft preparation, C.L. and F.C.; writing—review and editing, Y.C., H.W. and L.D.; project administration, L.D.; funding acquisition, L.D. All authors have read and agreed to the published version of the manuscript.

**Funding:** This research was supported by the National Science Foundation of China (NSFC: grants: 51901235, 51931005, 11790292, 11572324), the NSFC Basic Science Center Program for "Multiscale Problems in Nonlinear Mechanics" (Grant No. 11988102), Ye Qisun Science Foundation of National Natural Science Foundation of China (No. U2141204), the opening project of State Key Laboratory of Explosion Science and Technology (Beijing Institute of Technology, No. KFJJ18-14M).

**Data Availability Statement:** Available from the corresponding author upon request.

**Conflicts of Interest:** The authors declare no conflict of interest.

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
