# Peer review of "Crystal Plasticity Model Analysis of the Effect of Short-Range Order on Strength-Plasticity of Medium Entropy Alloys"

_metals, doi:10.3390/met12101757_

Round 1
Reviewer 1 Report
The authors have added a perturbation term to the resolved shear stress on a slip system to represent an effect of short range order on mechanical and deformation properties in a medium entropy alloy. The addition of this term allows for matching to experimental values of stress and strain. From this point of model validation, the authors are able to describe localized effects responsible for the observed macroscopic properties.
1. There are a few areas where the paper should provide more information to the reader. Namely, the authors should provide a physical description of why chemical short-range order is treated as a linear term in the shear equation. This would then lead into a discussion as to why it needs to be a floating parameter that changes for each of the three experiments that were compared to. (i.e. a discussion of why does the complexity of deformation described in atomic-level simulations boil down to a simple continuum parameter and how does this parameter relate to the atoms and the crystal's configurational entropy?)
2. The authors should provide references for all equations that were not explicitly derived in this work
3. The authors should provide references or explanations for their choices of material parameters in Table 1 given that quantitative values of Tau(SRO) are likely sensitive to the choices made here. Additionally, this also affects the quantitative description of maximum elongation later in the paper.
4. The authors should explain how they decided on the simulated grain structure in Figure 1(a)
5. The colon to comma to semicolon construction starting at line 148 is a bit of a grammatical mess. There appears to be a missing word in the sentence on line 244 "when the local __________ rotate more"
Reviewer 2 Report
Dear Authors,
In order to improve the quality of the article, I ask the authors to pay attention to the following:
1. In several places in the article, the authors mention the terms yield strength or yield stress. Since this stress is not expressed in the considered alloy, it is actually the proof strength. Please correct.
2. Grouping of literature is not a preferred way of citation. The number of cited reference must be separated from the preceding text. Please correct.
3. In many places in the text, the measurement unit should be separated from the value.
4. Eq. (7): Replace bold font with regular font. The explanation of the “sa” symbol is missing. Please correct.
5. Eq. (11): The authors are asked to check the correctness of the symbol “Ft” in this equation.
6. Line 187: Replace the point (between symbols) with a comma.
7. Eq. (13): The explanation of the symbols "k" and "damni" should be followed immediately after this equation.
8. Eq. (15): Does this equation contain the symbol "t" or "h" (thickness or width of twin lamellas). In addition, the meaning of the symbol "w" is not clear. Please correct.
9. Please check the correctness of the marked symbols:
10. The explanation of the symbol "ptn" should be followed immediately after Eq. (18).
11. Explanation of symbol "bp" in equations (21) and (24) is missing. Please correct.
12. Eq. (22): Replace bold font with regular font. Explanation of symbol "stwb" is missing. Please correct.
13. Lines from 151 to 153: The authors need to define more clearly the terms macro-stress and local stress.
14. Fig. 1b), 2a) 2b) and 2c): Please correct the symbol for dislocation resistance of SRO. Separate the measurement unit both from the value or the name of axis.
15. Table 1: Please separate the measurement unit from the value; Separate the values of interaction coefficient between slip systems; The cross-slip volume value would be better to express in nm. The simulations was performed with different values of dislocation resistance of SRO: from 0,10 to 60 MPa (not only 30 MPa). Please correct.
16. Line 170: An explanation of the abbreviation for TWIP steel is missing. Please specify.
17. Line 173: Incorrectly specified equation number for fitting parameter "A". Please correct.
18. Chapter 3.2.: Comparison with the experimental results of Miao et al. should also be mentioned.
19. Line 209: Replace the point with a comma (“… respectively. the local enlargements for the head …”).
20. Please separate the text in lines: 209 (“… end(a2) …” ), 211 (“… and(b2) …” ), 214 (“… 0.1,0.3. …”), 215 (“… 0.1,0.3. …”), 216 (“… 0.1,0.3. …”), 240 (“… under(a1,a2) …”), 245 (“… 5(a1 …”) and 247 (“… 5(a2 …”)
21. Fig. 3, 4 and 5: Please separate the measurement unit from the value.
22. Line 227: Correct the symbol of degree.
23. Line 240: Pleasce correct: “Mises stress and dislocation density distribution under” → “Mises stress and dislocation density distribution next to” .
Reviewer
Round 2
Reviewer 1 Report
Thank you for addressing my comments. The improved clarity in the authors' methodology has really strengthened the manuscript.
Author Response
We have optimized a few spellings and expressions in the revised manuscript.
Reviewer 2 Report
Dear Authors,
Thanks to the authors for the corrections and changes that contributed to the higher quality of the article. However, you should also pay attention to the following:
1. Eq. (7) and (22): Replace bold font with regular font.
2. Table 1: Separate the values of exponent in slip velocity. The description of this physical quantity must begin with a capital letter. Please correct.
3. Line 193: The number of cited reference must be separated from the preceding text.
4. Lines 204, 262: Please separate the measurement unit from the value.
5. Line 236: Correct the point in the middle of the sentence (“…respectively. the local enlargements…”)
6. Line 255: Correct the symbol of degree (º → °)
Reviewer
